# Genomic and Phenotypic Characterization of a Drug-Susceptible *Acinetobacter baumannii* Reveals Increased Virulence-Linked Traits and Stress Tolerance

**DOI:** 10.3390/biology14091201

**Published:** 2025-09-05

**Authors:** Wuen Ee Foong, Wenjun He, Xinxin Xiang, Jiabin Huang, Heng-Keat Tam

**Affiliations:** 1Department of Biochemistry and Molecular Biology, Hengyang Medical School, University of South China, Hengyang 421001, China; wuenee@hotmail.com (W.E.F.);; 2Department of Medical Microbiology, Hunan Provincial Key Laboratory for Special Pathogens Prevention and Control, Hengyang Medical School, University of South China, Hengyang 421001, China; 3National Health Commission Key Laboratory of Birth Defect Research and Prevention, Hunan Provincial Maternal and Child Health Care Hospital, Changsha 410008, China; 4Institute of Medical Microbiology, Virology and Hygiene, University Medical Center Hamburg-Eppendorf, Martinistraße 52, 20246 Hamburg, Germany

**Keywords:** *Acinetobacter baumannii*, heme utilization, virulence, serum resistance, desiccation tolerance

## Abstract

**Simple Summary:**

*Acinetobacter baumannii* is often recognized for its resistance to antibiotics, posing a serious threat in healthcare environments. In this study, we investigated a clinical isolate, HKAB-1, that despite being highly sensitive to many antibiotics, exhibits pronounced virulence-associated traits. Compared to the reference strain ATCC 19606, HKAB-1 demonstrates enhanced survival in serum and under desiccating conditions. HKAB-1 also forms robust biofilm and displays greater motility, phenotypes associated with persistence and pathogenicity. Genomic and transcriptomic analyses revealed that HKAB-1 harbours active iron acquisition and heme utilization systems, which are highly responsive to host-like conditions. Moreover, we found that genes associated with biofilm formation were highly induced in biofilm-forming cells. Conversely, the expression of *adeB* was markedly reduced, potentially explaining its antibiotic susceptibility despite harbouring multiple resistance genes. These findings reflect a potential evolutionary trade-off in certain *A. baumannii* strains, favoring virulence-associated traits over the expression of antimicrobial resistance mechanisms. This result highlights the need for continuous genomic surveillance to monitor emerging virulent but drug-susceptible strains like HKAB-1, which may serve as reservoirs for resistance development under selective pressures.

**Abstract:**

*Acinetobacter baumannii* is an opportunistic pathogen notable for multidrug resistance and environmental persistence. We characterized a clinical isolate, HKAB-1, which exhibits pronounced virulence-associated traits despite being highly susceptible to all tested antibiotics. HKAB-1 exhibited superior growth in MH2B, serum and desiccating conditions, robust biofilm formation, and active motility. Whole-genome sequencing identified two heme utilization clusters, multiple siderophore biosynthesis pathways, and other virulence-associated genes. Gene expression analysis revealed significant upregulation of heme utilization and siderophore biosynthetic gene clusters under serum exposure, indicating activation of iron uptake pathways under host-like conditions. Biofilm-associated genes, including *bap*, PNAG biosynthetic genes, and type IV pili components, were notably upregulated in biofilm-forming cells, supporting their role in driving the enhanced biofilm phenotype. Conversely, *adeB*, encoding a major RND efflux pump, was markedly downregulated, potentially explaining its drug-susceptible phenotype. Comparative genomic analysis highlighted differences in genes related to nutrient transport, metabolic pathways, and membrane biogenesis that may underpin its enhanced growth. These findings point to a potential trade-off between antibiotic resistance and virulence, underscoring the importance of monitoring antibiotic-susceptible yet highly virulent *A. baumannii* isolates as potential reservoirs for resistance evolution. Further investigation is warranted to elucidate the mechanisms underlying this phenotypic balance.

## 1. Introduction

*Acinetobacter baumannii* is the most commonly isolated species of its genus in clinical settings and has become a model organism for studying multidrug resistance (MDR) [1]. In recent years, this pathogen has posed an escalating threat to global healthcare systems; the pathogen causes a wide range of severe nosocomial infections, including those of the respiratory tract, skin, urinary tract, wounds, and bloodstream [2]. The persistence of this bacterium in healthcare environments is largely attributed to its remarkable ability to withstand harsh conditions such as desiccation, disinfectants, and prolonged exposure to antimicrobials [3]. The genome plasticity of *A. baumannii* plays a central role in its exceptional resilience, enabling rapid adaptation to environmental pressures through the acquisition and horizontal transfer of exogenous genetic material [4]. The ability to integrate mobile genetic elements such as insertion sequences, transposons, and resistance islands further promotes genome rearrangement and the stable integration of resistance determinants [5,6]. Alarming trends over the past decade reveal a growing prevalence of pandrug-resistant *A. baumannii* strains [7], underscoring the urgent need to understand the genetic and physiological basis of its survival and pathogenicity.

Among various resistance mechanisms, overexpression of efflux pumps and reduced outer membrane permeability play pivotal roles in conferring antibiotic resistance in *A. baumannii* [8,9]. Of particular interest, *A. baumannii* intrinsically encodes a large repertoire of multidrug efflux pumps, which are categorized into single-component transporters and tripartite systems [10]. These efflux systems, acting independently or in synergy, actively extrude a wide variety of structurally and chemically diverse substrates, including antibiotics, biocides, dyes, and detergents, leading to reduced intracellular drug accumulation and elevated minimum inhibitory concentrations (MICs) [11,12].

Beyond antimicrobial resistance, the ability of *A. baumannii* to form biofilms on both biotic and abiotic surfaces, as well as its capacity for surface-associated motility, also contributes significantly to its success as a nosocomial pathogen [13,14]. These traits are particularly important in the colonization of medical devices and invasive procedures, facilitating persistent hospital-acquired infections [15]. Moreover, both biofilm formation and motility are thought to promote the dissemination and acquisition of resistance genes, further enhancing the organism’s adaptability and persistence in hospital environments [16,17]. Collectively, the interplay of biofilm formation, motility, and multidrug resistance underpins the remarkable ability of *A. baumannii* to persist and spread within clinical settings.

In this study, we characterized a clinical *A. baumannii* isolate, HKAB-1, which displays accelerated growth kinetics and pronounced virulence-associated phenotypes, such as increased biofilm formation, motility, desiccation tolerance, and serum resistance, despite being broadly susceptible to antibiotics. Whole-genome sequencing identified HKAB-1 as sequence type ST392 and revealed the presence of multiple putative virulence factors, including the *hemO* gene cluster and heme utilization cluster 1. Comparative genomic analysis highlighted differences in genes involved in nutrient transport, metabolic pathways, and membrane biogenesis that may contribute to enhanced growth in HKAB-1. Notably, the expression of *adeB* was significantly repressed in HKAB-1, potentially accounting for its drug-susceptible phenotype despite harbouring multiple resistance determinants. Furthermore, under biofilm conditions, genes encoding the biofilm-associated protein Bap, polysaccharide poly-*N*-acetylglucosamine (PNAG) biosynthetic genes and type IV pili (T4P) components were highly induced, supporting their role in HKAB-1 robust biofilm phenotype. These findings point to a potential evolutionary trade-off between antibiotic resistance and virulence, underscoring the need to re-evaluate the clinical significance of antibiotic-susceptible *A. baumannii* in infection control and resistance surveillance. Further studies are warranted to elucidate the molecular mechanisms underpinning this phenotype balance.

## 2. Methods and Materials

### 2.1. Sample Collection and Bacterial Isolation

The sputum sample from a patient with a left-sided heart failure suspected to have a bacterial infection was collected for bacterial remuneration analysis. The samples were spread on blood agar and incubated at 37 °C overnight. The pure cultures were stored at −80 °C till further processing.

### 2.2. Growth Curve Measurement in MH2B

Colonies of *A. baumannii* were grown in Mueller Hinton II Broth (MH2B) liquid medium (Solarbio Science & Technology Co., Ltd., Beijing, China) at 37 °C for 16–18 h. Cultures were inoculated into fresh 120 mL MH2B medium at an OD_600_ of 0.05 and cultured at 37 °C with shaking at 150 rpm. The readout of OD_600_ was used to plot the bacterial growth curves by the “baranyi_without_lag” model using the “nlsMicrobio” package in R 4.3.3 (R Foundation, Vienna, Austria) [18,19].

### 2.3. Growth Curve Analysis in Bovine Serum Albumin

Overnight cultures were washed twice with 0.85% NaCl and inoculated into 150 μL of either MH2B liquid medium or 100% bovine serum albumin (Bio-Channel Biotechnology Co., Ltd., Nanjing, China) in a polystyrene, U-bottom 96-well plate (Chengdu Anqite Medical Co., Ltd., Chengdu, China) at an initial OD_600_ of 0.004. OD_600_ was recorded every 20 min for 16 h at 37 °C with agitation (linear amplitude: 2 mm) using Tecan Infinite M200 Pro Microplate Reader (Tecan, Männedorf, Switzerland). Growth curves were modelled using either the modified Gompertz model “gompertzm” or the Baranyi and Roberts model without a maximum population parameter “baranyi_without_Nmax” via the “nlsMicrobio” package in R 4.3.3 (R Foundation, Vienna, Austria) [18,19].

### 2.4. Measurement of Minimal Inhibition Concentration

Measurement of minimal inhibition concentration was performed as previously described [20]. Briefly, overnight cell cultures of *A. baumannii* strains were diluted to OD_600_ of 0.02 in fresh MH2B liquid medium, and 30 μL cultures were inoculated into 120 μL of serial 2-fold substrate dilutions in MH2B liquid medium in a polystyrene, U-bottom 96-well plate. The microtiter plates were incubated at 37 °C for 16 h with 180 rpm and the OD_600_ was Tecan Infinite M200 Pro Microplate Reader (Tecan, Männedorf, Switzerland). The MIC was defined as the lowest antibiotic concentration at which the OD_600_ value was below 0.1.

### 2.5. Biofilm Formation Assay

Biofilm formation assay was determined as previously described with slight modification [21]. Briefly, overnight cell cultures of *A. baumannii* strains were inoculated into 5 mL polystyrene tubes containing 1 mL of LB medium at an initial OD_600_ of 0.05 and statically incubated at 30 °C or 37 °C for 24 h. Bacterial growth in the liquid culture was determined by OD_600_. Subsequently, the supernatant of bacterial cultures was discarded, and the biofilm cells were rinsed with distilled water three times and then stained with 0.1% crystal violet for 20 min at room temperature. The crystal violet staining solution was discarded, the tubes were again rinsed three times with distilled water, and the stained biofilm cells were solubilized with absolute ethanol. The solubilized biofilm cells were quantified at OD_595_. Biofilm formation was expressed as the OD_595_/OD_600_ ratio to normalize the total bacterial growth.

### 2.6. Motility Assays

Both swarming and twitching motility assays were determined as previously described with slight modification [22]. For swarming assay, overnight cultures were diluted to OD_600_ of 0.1, and a 2 μL drop was pipetted onto an LB medium consisting of 0.4% agar. The agar plates were incubated at 37 °C for 24 h. For twitching assay, overnight cultures were diluted to OD_600_ of 0.1, and a 2 μL drop of cultures was stab-inoculated through the LB medium consisting of 0.8% agar to form an interstitial colony between the petri dish and the agar medium. The agar plates were incubated at 37 °C for 24 h. After incubation, the agar was discarded, and the plates were stained with 0.2% crystal violet before visualization.

### 2.7. Desiccation Assay

Desiccation assay was assessed as previously described with slight modification [23]. Overnight cultures were grown in MH2B medium at 37 °C and then harvested and washed twice with 0.85% NaCl. The cell suspension was adjusted to an OD_600_ of 2.0 in the same buffer. Aliquots of 20 μL were pipetted onto cellulose acetate membrane filters (Hangzhou Special Paper Industry Co., Ltd., Hangzhou, China) (0.45 μm pore size) and air-dried under laminar flow. The dried membranes were then placed in uncovered petri dish inside airtight Glasslock containers (17.7 × 13.1 × 6.8 cm) (SGC Solutions Co., Ltd., Seoul, Republic of Korea) containing 30 g of Drierite desiccant (W A Hammond Drierite Co., Ltd., Xenia, OH, USA) to maintain a controlled relative humidity of 20 ± 3%. The containers were incubated at 24 °C. For viable cell counts at day 0, membranes were transferred into 2 mL tubes containing 1 mL of 0.85% NaCl and gently vortexed for 5 min at room temperature. Serial dilutions were plated on LB agar to determine colony-forming units (CFUs).

### 2.8. Hemolysis and Protease Assays

Hemolysis assay was performed on a Columbia blood agar plate supplemented with 5% defibrinated horse blood (Chongqing Pangtong Medical Devices Co., Ltd., Chongqing, China) as previously described [24]. Briefly, 10 μL of overnight cultures adjusted to an OD_600_ of 2.5 were inoculated onto the blood agar plate and incubated at 37 °C for 48 h. Protease assay was conducted on skim milk agar plates (Shandong Tuopu Biol-Engineering Co., Ltd., Zhaoyuan, China), which were incubated at 37 °C for 24 h.

### 2.9. RNA Extraction

To evaluate the expression of virulence-associated and RND efflux pump genes, overnight cultures of *A. baumannii* strains were diluted to an initial OD_600_ of 0.05 in 25 mL of LB medium and incubated at 37 °C with shaking at 180 rpm until reaching an OD_600_ of 0.5–0.7. To assess the expression of genes associated with serum tolerance, *A. baumannii* cells grown in serum-containing medium and MH2B (as the control) were harvested at an OD_600_ of 0.5–0.7. A 500 μL aliquot of each culture was stabilized using RNAstore Reagent (Tiangen Biotech Co., Ltd., Beijing, China). Total RNA was extracted using the RNAprep Pure Bacteria Kit (Tiangen Biotech Co., Ltd., Beijing, China). To determine the expression of biofilm-associated genes, biofilms from four technical replicates of the same biological sample were pooled and resuspended in RNAprotect Bacteria Reagent (Qiagen, Hilden, Germany). RNA extraction was conducted using the RNeasy Mini Kit (Qiagen, Hilden, Germany). The concentration and purity of RNA were determined using TGem Pro spectrophotometer (Tiangen Biotech Co., Ltd., Beijing, China).

### 2.10. Reverse Transcription qPCR

Reverse transcription was performed using the FastKing gDNA Dispelling RT SuperMix II (Tiangen Biotech Co., Ltd., Beijing, China) with 500 ng of total RNA as template. Quantitative PCR (qPCR) was conducted on an Applied Biosystems StepOnePlus (Applied Biosystems, Foster City, CA, USA) or Bio-Rad CFX Connect (Bio-Rad Laboratories, Hercules, CA, USA) thermal cycler using 2× Universal SYBR Green Fast qPCR Mix (AbClonal Technology, Woburn, MA, USA) with 400 ng of total cDNA per reaction. Primer sequences are listed in Appendix A, with *rpoB* used as the reference gene. Gene expression analysis was conducted as previously described [25]. ΔC_T_ values were calculated as C_T(rpoB)_ − C_T(target gene)_. Differences in mean ΔC_T_ values between “treatment” (e.g., serum, biofilm, or HKAB-1) and “control” groups (e.g., LB or ATCC 19606) were analyzed using an analysis of variance (ANOVA) model implemented in R version 4.4.1 (R Foundation, Vienna, Austria) [19], incorporating Gene and Gene:Treatment interaction term. Tukey’s honestly significant difference (HSD) test was applied to determine statistically significant differences in gene expression between groups.

### 2.11. Extraction of Genomic DNA and Genome Sequencing

The genomic DNA was sheared to an average size of 350 bp using the Covaris LE220R-plus system (Covaris, Woburn, MA, USA) and then end-polished, A-tailed, and ligated to full-length Illumina adapters. The constructed libraries were sequenced on Illumina platforms (Illumina, San Diego, CA, USA) with 2 × 150 bp paired-end reads by Novogene Bioinformatics Technology Co., Ltd. (Beijing, China). Quality control and trimming of paired-end reads were performed using Trimmomatic v0.39 (Usadel Lab, RWTH Aachen University, Aachen, Germany) [26]. De novo assembly was performed using SPAdes v3.15.5 (Algorithmic Biology Laboratory, St. Petersburg Academic University, Russian Academy of Sciences, St. Petersburg, Russia) [27], resulting in 72 contigs. Contig annotation was carried out using the Bacterial and Viral Bioinformatics Resource Center (BV-BRC) pipeline [28]. All contigs equal to or longer than 500 bp were successfully scaffolded using Multi-CAR (Algorithm and Bioinformatics Laboratory, Department of Computer Science, National Tsing Hua University, Republic of China) [29], which employs multiple reference genomes to arrange and orient contigs, resulting in a chromosomal assembly. Average nucleotide identity (ANI) analysis was calculated using online ANI calculator (https://www.ezbiocloud.net/tools/ani, accessed on 30 May 2025), employing OrthoANI algorithm [30].

### 2.12. Genome Annotation and Analysis

Genomes were analyzed for loci associated with antimicrobial resistance using Abricate v.1.01 (https://github.com/tseemann/abricate, Seemann Lab, University of Melbourne, Melbourne, Australia, accessed on 6 April 2025), with reference to the MEGARes v.2.0 (Microbial Ecology Group; Colorado State University, Texas A&M University, University of Florida, University of Minnesota, West Texas A&M University, Canyon, TX, USA) and Antibiotic Resistance Gene-ANNOTation (ARG-ANNOT) v6 (Unité de Recherche sur les Maladies Infectieuses et Tropicales Emergentes, Faculté de Médecine et de Pharmacie, Aix-Marseille Université, Marseille, France) [31,32]. Virulence factors and prophage sequences of *A. baumannii* HKAB-1 were analyzed with VFDB v2022 (NHC Key Laboratory of Systems Biology of Pathogens, Institute of Pathogen Biology, Chinese Academy of Medical Sciences & Peking Union Medical College, Beijing, China) [33] and PHASTEST v1.0.1 (Wishart Lab, Department of Biological Sciences, University of Alberta, Edmonton, AB, Canada) [34], respectively. Genome map was created using Proksee v1.3.0 (Stothard Research Group, Agriculture, Food & Nutritional Science, University of Alberta, Edmonton, AB, Canada) [35]. Multilocus sequence typing (MLST) analysis was conducted using PubMLST (https://pubmlst.org, accessed on 30 May 2025) [36]. The capsular polysaccharide (CPS) locus (KL) and lipooligosaccharide outer core locus (OCL) of the HKAB-1 strain were identified using Kaptive Web v1.3.0 (Melbourne eResearch Group, University of Melbourne, Melbourne, Australia) with reference to the *A. baumannii* database [37]. The core-genome phylogenetic tree of *A. baumannii* HKAB-1 and representative *Acinetobacter* strains was constructed using annotated genome assemblies (GFF3 format) processed with Roary v3.13.0 (Pathogen Genomics, The Wellcome Trust Sanger Institute, Wellcome Trust Genome Campus, Hinxton, Cambridge, UK) [38] to identify core genes. Multiple sequence alignment was performed using MAFFT v7.526 (Immunology Frontier Research Center, Osaka University, Osaka, Japan) [39], and phylogenetic inference was carried out with RAxML-NG v1.2.2 (Computational Molecular Evolution Group, Heidelberg Institute for Theoretical Studies, Heidelberg, Germany) [40] under the GTR + G model with 1000 bootstrap replicates.

## 3. Results

### 3.1. Strain Collection and Identification

A patient with left-sided heart failure and clinical signs consistent with a bacterial respiratory infection was admitted to the hospital. A bacterial isolate, designated HKAB-1, was recovered from the patient’s sputum following culturing on a Columbia blood agar plate supplemented with 5% defibrinated horse blood. No hemolytic was observed on the agar plate (Appendix A). Based on preliminary phenotypic characteristics, the isolate was presumptively identified as *A. baumannii*. The patient exhibited multiple high-risk factors, including older age, severe cardiac dysfunction, and multi-organ dysfunction. In accordance with hospital treatment protocols, a seven-day course of piperacillin–tazobactam was initiated as empiric antimicrobial therapy. Following completion of the antibiotic regimen, the patient showed marked improvement in respiratory symptoms. By the ninth day of hospitalization, cardiac examination revealed normal heart sounds with no pathological murmurs across all valve areas, while pulmonary auscultation detected a few dry rales in both lungs. Given the clinical improvement, the patient was deemed fit for discharge. The isolate was subsequently referred to the Department of Medical Microbiology, Hengyang Medical School, University of South China, for molecular identification and phenotypic characterization.

### 3.2. HKAB-1 Is More Susceptible to a Broad Range of Antibiotics

In recent years, infections associated with *A. baumannii* have emerged as a significant concern in clinical settings, primarily due to their multidrug resistance phenotype [1]. Therefore, the antibiotic susceptibility profile of the clinical isolate HKAB-1 was evaluated relative to the laboratory strain ATCC 19606 to delineate strain-specific resistance patterns. Unexpectedly, HKAB-1 displayed greater susceptibility to most of the tested antibiotics compared to the reference strain ATCC 19606, with the exception of oxacillin (2-fold difference, MIC: 512 μg/mL) and rifampicin (2-fold difference, MIC: 4 μg/mL) (Table 1). Notably, HKAB-1 demonstrated significantly enhanced susceptibility, with 4- to 8-fold lower MIC for amikacin, azithromycin, ciprofloxacin, doripenem, tetracyclines, and tobramycin and up to 16-fold reduction in MIC for amoxicillin and tigecycline (Table 1).

### 3.3. HKAB-1 Exhibits Enhanced Biofilm Formation and Serum Resistance

The persistence of *A. baumannii* in clinical settings is largely attributed to its ability to form biofilm, which is a key virulence factor in infections [41,42]. To investigate these traits, we evaluated the capability to form biofilm and serum resistance of the HKAB-1 strain. To ensure accurate normalization in the biofilm assay, we first analyzed the growth kinetics of the clinical isolate HKAB-1 and the laboratory strain ATCC 19606 in MH2B medium. The HKAB-1 strain exhibited a higher growth rate in MH2B compared to ATCC 19606, with maximum specific growth rates (μ_max_) of 0.87 ± 0.03 h^−1^ and 0.63 ± 0.02 h^−1^ (unpaired Student’s *t*-test, *p* < 0.005), respectively (Figure 1A,B). Consistently, OD_600_ measurements at the 5 h time point were also higher for HKAB-1, with mean OD_600_ values of 1.57 ± 0.07 for ATCC 19606 and 1.91 ± 0.07 for HKAB-1 (unpaired Student’s *t*-test, *p* < 0.05) (Figure 1C).

Given the significant differences in growth dynamics, biofilm quantification using crystal violet staining was normalized to the OD_600_ of the corresponding statically incubated planktonic cultures to ensure comparability and accuracy. Notably, HKAB-1 formed slightly more biofilm at 37 °C than at 30 °C, although the difference was not statistically significant (unpaired Student’s *t*-test, *p* = 0.11), whereas ATCC 19606 showed no temperature-dependent variation in biofilm formation (Figure 1E). Interestingly, HKAB-1 demonstrated a significantly greater capacity for biofilm formation than ATCC 19606 at 30 °C (unpaired Student’s *t*-test with Welch’s correction, *p* < 0.05), whereas no notable difference was observed between the strains at 37 °C (Figure 1E).

Clinical *Acinetobacter* spp are known to exhibit high resistance to serum [43]. To assess the survival of the HKAB-1 strain in bovine serum albumin (BSA), we monitored the growth of both *A. baumannii* strains in BSA and MH2B using a polystyrene, U-bottom 96-well plate. As anticipated, HKAB-1 showed enhanced growth in MH2B compared to ATCC 19606 (Figure 1A,D), with mean OD_600_ values at the 16 h time point of 1.03 ± 0.01 for HKAB-1 and 0.85 ± 0.01 for ATCC 19606 (unpaired Student’s *t*-test, *p* < 0.005) (Figure 1D). Remarkably, HKAB-1 also demonstrated robust growth in 100% BSA, whereas ATCC 19606 was highly susceptible to BSA, with mean OD_600_ values of 0.57 ± 0.01 and 0.19 ± 0.02, respectively (unpaired Student’s *t*-test, *p* < 0.005) (Figure 1D). These findings highlight the enhanced adaptability of HKAB-1 under both nutrient-rich and serum-based conditions (Figure 1).

### 3.4. HKAB-1 Exhibits Enhanced Motility

Surface-associated motility has been recognized as a common trait among clinical isolates of *A. baumannii* [13]. To evaluate the motility of clinical isolate HKAB-1, we determined the swarming and twitching motility of both HKAB-1 and ATCC 19606 on semi-solid agar. After 24 h of incubation, HKAB-1 exhibited markedly greater swarming motility on the surface of agar and twitching motility at the agar–plastic interphase compared to the laboratory strain, with mean diameter of 2.8 ± 0.1 cm and 2.1 ± 0.1 cm (unpaired Student’s *t*-test, *p* < 0.005), respectively (Figure 2A–D). Congruent to previous findings [44], ATCC 19606 showed no motility on semi-solid media (Figure 2A–D).

### 3.5. HKAB-1 Tolerates Desiccation

Previous studies have shown that biofilm formation significantly enhances desiccation tolerance in *A. baumannii*, albeit potentially at the cost of reduced antibiotic resistance [45]. We hypothesized that the high antibiotic susceptibility and enhanced biofilm-forming ability of the HKAB-1 strain might confer improved tolerance to prolonged desiccation. Consistent with this hypothesis, desiccation resulted in a 66.3% reduction in viable HKAB-1 cells compared to the initial inoculum (Tukey’s HSD test, *p* < 0.005), whereas the viability of ATCC 19606 dropped to nearly undetectable levels after just one day of incubation (Figure 2E). Notably, after four days of desiccation, HKAB-1 retained approximately 15.4% viability (Tukey’s HSD test, *p* < 0.005) (Figure 2E), underscoring its enhanced desiccation resistance.

### 3.6. Genomic Context of A. baumannii HKAB-1 Strain

To explain the differences in antibiotic sensitivity between the clinical isolate and the ATCC 19606 strain, whole-genome sequencing of HKAB-1 was performed, and the average nucleotide identity (ANI) analysis [30] revealed that the isolate shared 97.88% identity with *A. baumannii* (NCBI accession CP059040), confirming species-level classification. The assembled genome has a total length of 3,758,367 bp and consists of a single circular chromosome (Figure 3A). Further validation of species identity and phylogenetic relationships was performed through core-genome analysis incorporating publicly available *Acinetobacter* genomes. Indeed, HKAB-1 clusters within the *A. baumannii* clade, demonstrating close evolutionary relatedness to clinically relevant strains such as AB030 (Figure 3B). Notably, no detectable plasmids were detected. The overall G + C content is 38.92%. Genome annotation predicted a total of 3559 genes, including 3488 coding sequences (CDSs) and 71 RNA genes, comprising 4 rRNAs and 63 tRNAs (Table 2). Among the 3488 predicted CDSs, 3446 protein-coding genes were assigned to 21 functional categories based on the COG database (Appendix A). Moreover, a considerable number of genes were categorized as unknown function (19.47%), suggesting that further analysis is needed to elucidate their functions. In silico MLST identified HKAB-1 as belonging to sequence type ST392 (Appendix A) according to the Pasteur scheme [46]. Notably, ST392 has only been reported once previously, from a single isolate in Taiwan in 2010, raising questions about its geographic distribution and potential emergence.

### 3.7. Comparative Analysis of Genes Associated with Antimicrobial Resistance

Antimicrobial resistance gene profiling using ABRicate (https://github.com/tseemann/abricate, accessed on 6 April 2025), with reference to the MEGARes 2.0 and Antibiotic Resistance Gene-ANNOTation (ARG-ANNOT), identified a total of 27 and 28 open reading frames (ORFs) putatively associated with resistance genes in the HKAB-1 and ATCC 19606 strains (Table 3) [31,32]. Of the 27 resistance genes, 21 are associated with the efflux pumps, including 10 ORFs from the resistance nodulation cell division (RND) family, 4 ORFs encoding putative major facilitator superfamily (MFS) family, 2 ORFs from the multidrug and toxic efflux (MATE) family, 1 ORF from the small multidrug resistance (SMR) family, and 5 ORFs encoding putative transcriptional regulators. The remaining 6 ORFs are associated with antibiotic inactivation, comprising 5 β-lactamase encoding genes and 1 gene encoding an aminoglycoside nucleotidyltransferase (*ant(3*″*)-IIa*). Similar to ATCC 19606, HKAB-1 lacks the gene encoding outer membrane protein *adeC* gene in its genome. The absence of *adeC* from the *adeAB* loci in *Acinetobacter* genome is unprecedented, as AdeAB could recruit another outer membrane protein AdeK to form a functional tripartite complex [47]. Interestingly, HKAB-1 harbours a comparable number of antimicrobial resistance genes to ATCC 19606 but exhibits increased susceptibility to all tested antibiotics.

### 3.8. Comparative Analysis of Genes Associated with Virulence Factors

Since the HKAB-1 strain exhibited higher virulence, including higher biofilm formation, motility, and serum resistance, compared to ATCC 19606, we evaluated the genetic background of different virulence traits between HKAB-1 and ATCC 19606 strains. The HKAB-1 genome encodes 106 ORFs associated with virulence factors, which are categorized into seven major functional families comprising adherence, biofilm formation, phospholipases, immune evasion, iron uptake, regulatory functions, and serum resistance (Appendix A). In comparison, the ATCC 19606 genome contains 95 virulence-associated ORFs, lacking key heme utilization genes, including the *hemO* gene cluster (ACP71R_11060–ACP71R_11105) and heme utilization cluster 1 (ACP71R_06965–ACP71R_07025), both of which are present in HKAB-1 (Appendix A). Additionally, the genome also harbours two putative prophage regions with high sequence similarity to *PHAGE_Acinet_Bphi_B1251_NC_019541* (Figure 3A). The first prophage is an intact prophage spanning 43.7 Kb (positions 2,159,893 to 2,203,690 bp) with a GC content of 38.67%. The second prophage is an incomplete prophage, 30.9 Kb in length, located between positions 1,930,574 and 1,961,551 bp, with a GC content of 35.65%.

In addition to virulence factors identified through the VFDB, the capsular polysaccharide (KL) locus is recognized as a critical determinant of *A. baumannii* pathogenicity [48]. The HKAB-1 strain carries the KL107 locus, located between the ACP71R_14980 encoding a putative FKBP-type peptidyl-prolyl cis-trans isomerase (*fkpA*) and *lldP* genes, and on the one hand, the OCL3 locus, situated between ACP71R_00345 (encoding a hypothetical protein) and the ACP71R_00295 encoding putative branched-chain amino acid transaminase (*ilvE*) genes. These loci are responsible for the biosynthesis and export of capsular polysaccharide (CPS) and outer core locus (OCL), respectively. Notably, OCL3 of the HKAB-1 strain differs from the more common OCL configurations, which typically reside between *ilvE* and *aspS* [49].

### 3.9. Relative Expression of Efflux Pump and Virulence-Associated Genes

To investigate the functional relevance of efflux pumps and virulence-associated genes in antibiotic resistance and pathogenicity, we quantified the expression levels of these genes under relevant phenotypic conditions using RT-qPCR. No significant differences were detected in the transcription of *adeG* and *adeJ* between HKAB-1 (*adeG*: ∆C_T_ = −9.45 ± 0.15; *adeJ*: ∆C_T_ = −1.97 ± 0.15) and ATCC 19606 (*adeG*: ∆C_T_ = −9.42 ± 0.15; *adeJ*: ∆C_T_ = −1.49 ± 0.15) in LB medium (Figure 4A, see x-axis). In contrast, *adeB* was markedly downregulated in HKAB-1 (ΔΔC_T_ = −7.27 ± 0.21; fold change = 0.0065), potentially contributing to its antibiotic-susceptible phenotype despite harbouring multiple resistance genes.

Given the roles of biofilm-associated protein Bap, Csu pili, PNAG biosynthesis, and T4P in biofilm formation and twitching motility (Appendix A) [50,51,52,53], we examined their transcriptional profiles in HKAB-1. ATCC 19606 exhibited relatively low expression, except for *csuC* (*bap*: ∆C_T_ = −4.95 ± 0.32; *csuC*: ∆C_T_ = −1.22 ± 0.32; *pgaB*: ∆C_T_ = −7.58 ± 0.32; *pilA*: ∆C_T_ = −10.04 ± 0.32; *pilO*: ∆C_T_ = −8.17 ± 0.32) (Figure 4B, see x-axis). No appreciable transcriptional differences were observed for *csuC* encoding a chaperone involved in Csu pili assembly and *pgaB* encoding a deacetylase in PNAG biosynthesis across the two strains (Figure 4B). However, *bap* and T4P-specific genes, including *pilA* encoding a putative pilin and *pilO* encoding a T4P biogenesis protein, were significantly upregulated in HKAB-1 (*bap*: ΔΔC_T_ = 3.03 ± 0.45; fold change = 8.14; *pilA*: ΔΔC_T_ = 5.54 ± 0.45; fold change = 46.54; *pilO*: ΔΔC_T_ = 2.42 ± 0.45; fold change = 5.34) (Figure 4B), consistent with its robust biofilm and motility phenotypes.

To further assess the functional role of biofilm-associated genes in biofilm development, we compared their expression between biofilm-associated cells and planktonic cultures. Consistent with our earlier results (Figure 4B), *csuC* showed no appreciable transcriptional difference between the two conditions (Figure 4C). Unexpectedly, *pgaB*, which was expressed at low levels in planktonic cells (Figure 4B), was strongly induced in biofilm cells (ΔΔC_T_ = 5.55 ± 0.43; fold change = 46.79), supporting a key role for PNAG biosynthesis in biofilm formation (Figure 4C). As anticipated, *bap* and T4P-specific genes (*pilA* and *pilO*) were further upregulated in biofilm-associated cells compared to planktonic cells (*bap*: ΔΔC_T_ = 2.06 ± 0.43; fold change = 4.17; *pilA*: ΔΔC_T_ = 2.73 ± 0.43; fold change = 6.62; *pilO*: ΔΔC_T_ = 3.77 ± 0.43; fold change = 13.60) (Figure 4B,C). Intriguingly, both *adeB* and *adeG*, RND efflux pump genes expressed at relatively low levels in planktonic HKAB-1 (Figure 4A), were notably induced in biofilm-associated cells (*adeB*: ΔΔC_T_ = 7.88 ± 0.43; fold change = 236.19; *adeG*: ΔΔC_T_ = 7.54 ± 0.43; fold change = 185.98), whereas *adeJ* expression remained unchanged (Figure 4C). Given previous reports linking siderophore biosynthesis to biofilm development [54], we also examined siderophore-related genes and found that both *basA* (acinetobactin cluster) and *bfnL* (fimsbactin cluster) were strongly induced in biofilm cells (*basA*: ΔΔC_T_ = 6.05 ± 0.43; fold change = 66.43; *bfnL*: ΔΔC_T_ = 3.84 ± 0.43; fold change = 14.33) (Figure 4C), indicating a contributory role in biofilm development. Collectively, the pronounced upregulation of *bap* and T4P genes in HKAB-1 relative to ATCC 19606, together with their further induction in biofilm conditions and the concurrent activation of PNAG, RND efflux pumps, and siderophore systems, suggests a synergistic network of factors driving the robust biofilm phenotype of HKAB-1.

The heme utilization cluster plays a crucial role in bacterial survival under serum conditions [55]. Given that HKAB-1 harbours heme utilization genes, which are absent in ATCC 19606, we evaluated the regulatory response of iron acquisition systems to serum exposure by analyzing the expression of genes involved in heme uptake and siderophore biosynthesis. Under standard growth conditions in MH2B, expression of heme acquisition genes, including *hemO*, encoding a putative heme oxygenase and ACP71R_07005, a predicted TonB-dependent receptor from heme utilization cluster 1, along with siderophore biosynthesis genes such as *basA* and *bfnL*, remained relatively low (Figure 4D; see x-axis). Under exposure to serum, all four genes were significantly activated (*hemO*: ∆∆C_T_ = 9.13 ± 0.42; fold change = 558.50; ACP71R_07005: ∆∆C_T_ = 1.89 ± 0.42; fold change = 3.71; *bfnL*: ∆∆C_T_ = 6.05 ± 0.42; fold change = 66.29; *basA*: ∆∆C_T_ = 4.32 ± 0.42; fold change = 19.96) (Figure 4D), suggesting that iron acquisition pathways are strongly induced under host-mimicking conditions.

## 4. Discussion

The high antibiotic susceptibility phenotype of HKAB-1 is congruent with previous observations that certain *A. baumannii* and *Acinetobacter lwoffii* isolates, despite being susceptible to antibiotics, can still cause a broad range of infections [56]. Notably, while *A. lwoffii* is generally considered antibiotic-susceptible and carries relatively few resistance genes, it has recently emerged as a prevalent cause of infections, particularly in neonatal intensive care units, and is increasingly associated with MDR phenotypes [56,57]. These findings suggest that antibiotic susceptibility does not necessarily preclude the capacity for nosocomial outbreaks and that *Acinetobacter* spp. may still acquire resistance determinants under clinical selection pressure. Consistent with this observation, genome analysis of HKAB-1 revealed the presence of various antibiotic resistance genes, including those encoding clinically significant RND efflux pumps, which are key contributors to MDR phenotypes in *A. baumannii* [58]. Intriguingly, the transcript levels of *adeB* is relative low in HKAB-1 (∆C_T_ = −10.11 ± 0.15) compared to ATCC 19606 (∆C_T_ = −2.85 ± 0.15) (Figure 4A, see x-axis), and comparative gene analysis between HKAB-1 and ATCC 19606 indicates downregulation of *adeB* in HKAB-1 (ΔΔC_T_ = −7.27 ± 0.21; fold change = 0.0065) (Figure 4A). In contrast, no significant differences in expression levels of *adeG* and *adeJ* were observed under standard growth medium (Figure 4A). These findings suggest that reduced *adeB* expression is likely to account for the increased susceptibility of HKAB-1, as this efflux pump is critical in multidrug resistance in clinical *A. baumannii* isolates [8,59]. Moreover, reduced activity or downregulation of other resistance mechanisms, such as β-lactamases or non-RND efflux systems, may also contribute to the observed antibiotic susceptibility phenotype. Together, these findings indicate that although HKAB-1 currently remains antibiotic-susceptible, it harbours the genetic potential to express and activate resistance traits under appropriate selective pressures, warranting careful monitoring of such strains in clinical settings.

The ability of HKAB-1 to cause infection in the host led us to hypothesize that this strain may exhibit enhanced virulence phenotypes compared to the laboratory strain ATCC 19606. Consistent with this hypothesis, phenotypic characterization revealed that HKAB-1 demonstrates superior growth in serum and under desiccated conditions and the ability to form biofilm; it also displays both swarming and twitching motility (Figure 1 and Figure 2), while lacking hemolytic and proteolytic activities (Appendix A), traits collectively associated with virulence in *A. baumannii* [13,41,42,43,60]. Previous studies have suggested that the acquisition of MDR often incurs a fitness cost, including impaired growth and reduced tolerance to environmental stressors, while increased biofilm formation may offset antibiotic susceptibility in *A. baumannii* [45,61]. In line with these observations, HKAB-1 displays increased desiccation resistance and robust biofilm-forming capacity despite being highly susceptible to all tested antibiotics (Figure 1E and Figure 2E). Notably, HKAB-1 also exhibited higher metabolic capacity, as evidenced by faster growth kinetics and greater biomass yields (higher OD_600_ readings) in both MH2B and serum-based media (Figure 1A,D). Comparative analysis of annotated genes across categories of clusters of orthologous groups (COGs) revealed that HKAB-1 possesses more genes in several growth-related functions, including energy production and conversion (C), amino acid metabolism (E), inorganic ion transport (P), cell envelope biogenesis (M), and carbohydrate metabolism (G) (Appendix A). The enrichment of these COGs suggests an enhanced modulation of nutrient uptake, metabolic pathway genes, and membrane synthesis, which may collectively reduce physiological bottlenecks and enhance growth efficiency. Of particular note, we observed significant downregulation of the *adeB* efflux pump gene in HKAB-1 (Figure 4A). Given the energetic cost associated with resistance mechanisms [62], reduced *adeB* expression may alleviate metabolic burden, thereby contributing to improved growth. Together, these findings may support a model in which decreased antimicrobial resistance is offset by enhanced virulence traits and environmental fitness. The molecular mechanisms underlying this trade-off remain unclear and warrant further investigation.

Further genomic analysis revealed that HKAB-1 harbours the *hemO* gene cluster (ACP71R_11060–ACP71R_11105), as well as an additional heme utilization cluster 1 (ACP71R_06965–ACP71R_07025) (Appendix A). The *hemO* cluster, frequently found in MDR epidemic strains, has been implicated in increased virulence [55,63] and is thought to enhance bacterial survival in serum by facilitating iron acquisition from heme [55]. In agreement with these findings, we observed significant upregulation of *hemO* and ACP71R_07005, which encodes a heme oxygenase and a putative TonB-dependent receptor, respectively, in response to serum exposure (Figure 4D). Moreover, genes involved in the biosynthesis of the siderophores acinetobactin (*basA*) and fimsbactin (*bfnL*) were also markedly upregulated under these conditions (Figure 4D). These results indicate that both heme utilization and siderophore-mediated iron acquisition are critical for bacterial fitness in serum. We posit that the concurrent presence of *hemO* cluster, heme utilization cluster 1, and multiple siderophore systems contributes to the superior survival of HKAB-1 in serum, which contains limited free iron but is rich in heme-bound iron [64]. Although ATCC 19606 has been shown to utilize heme, earlier studies employed higher heme concentrations than those found in serum [65], potentially explaining its reduced growth under these conditions.

T4P is key virulence factor in *A. baumannii*, mediating host cell adhesion, biofilm formation, and twitching motility [53]. PilN and PilO are highly conserved components of the PilMNOP inner membrane alignment subcomplex, and the proper heterodimerization of PilNO is crucial for T4P biogenesis [66]. Comparative genomic analysis revealed that the ATCC 19606 strain harbours a disrupted *pilN* gene (Appendix A). The absence of a functional *pilN* likely compromises pilus assembly, contributing to its deficiencies in motility and biofilm formation. In contrast, all T4P in HKAB-1, including *pilN*, are intact (Appendix A), consistent with its retained twitching motility and robust biofilm phenotype. Moreover, prior studies suggest that variation in *pilA*, which encodes the major pilin subunit, can influence pilin surface electrostatics and modulate pilus-associated functions in *A. baumannii* [44,53]. PilA variants in *Acinetobacter* diverge into glycosylated forms, with a C-terminal serine residue serving as a glycosylation site and non-glycosylated forms lacking this residue (Appendix A). Glycosylation of PilA is mediated by a T4P-specific O-oligosaccharyltransferase *tfpO* located downstream of *pilA* [67]. While the functional impact of PilA glycosylation remains uncertain, strain-specific variants have been linked to distinct phenotypes [68]. For instance, the AB5075 variant is associated with twitching motility, whereas the ACICU variant promotes biofilm formation [53]. Phylogenetic analysis of PilA sequences from 19 *A. baumannii* strains revealed that HKAB-1 PilA forms a distinct clade, clustering more closely with variants from ACICU, NIPH-329, Ab44444, and OIFC137 than with AB5075, BIDMC57, or ATCC 19606 (Appendix A). Notably, despite clustering with ACICU, HKAB-1 PilA lacks the terminal serine residue, placing it within the non-glycosylated subgroup (Appendix A). These findings suggest a potential strain-specific role for HKAB-1 PilA in modulating motility and biofilm formation. In addition, elevated expression of *pilA* and *pilO* in HKAB-1 supports the enhanced twitching motility and biofilm phenotypes observed in this strain (Figure 1E, Figure 2A,B and Figure 4C).

Moreover, HKAB-1 encodes *bap*, a biofilm-associated protein essential for biofilm development in *A. baumannii* [50], which is also present in ATCC 19606. The markedly elevated expression of *bap* and T4P genes (e.g., *pilA* and *pilO*) in HKAB-1, coupled with their further induction under biofilm conditions and the upregulation of PNAG biosynthesis (Figure 4B,C), likely underpins the molecular basis of its enhanced biofilm-forming capacity. Intriguingly, both AdeB and AdeG RND efflux pumps, expressed at low levels in planktonic cultures, were highly responsive to biofilm conditions (Figure 4A,C), implicating that RND efflux pumps play roles in biofilm maturation or maintenance [69]. Biofilm formation, mediated in part by Bap and RND efflux systems, is a critical virulence mechanism that facilitates *A. baumannii* persistence under hostile host and environmental conditions [41,69,70]. Previous studies have further suggested that biofilm production is often correlated with both antibiotic susceptibility and serum resistance. Specifically, high biofilm-producing isolates often exhibit lower resistance to antibiotics in planktonic culture but display greater resistance to host factors such as serum, as well as enhanced resistance to desiccation [15,42,45,71]. Collectively, the biofilm-forming capacity of HKAB-1 may confer a selective advantage under host-associated stress or antibiotic exposure and potentially promote the acquisition or persistence of multidrug resistance within biofilm communities.

## 5. Conclusions

Although *A. baumannii* HKAB-1 harbours a wide array of antimicrobial resistance genes, including clinically relevant putative β-lactamases, as well as multiple RND and MFS efflux pumps, the strain remains susceptible to all tested antibiotics. Notably, genome analysis also reveals the presence of diverse virulence factors and prophage sequences, indicating a genomic architecture favorable to pathogenicity. Specifically, HKAB-1 harbours heme utilization clusters and multiple biofilm-associated genes, contributing to its capability to survive in serum and to its enhanced biofilm formation compared to ATCC 19606. These findings highlight the critical need for continuous surveillance of *A. baumannii* clinical isolates in order to prevent the emergence and spread of resistance in strains harbouring extensive multidrug resistance determinants.

## Figures and Tables

**Figure 1 biology-14-01201-f001:**
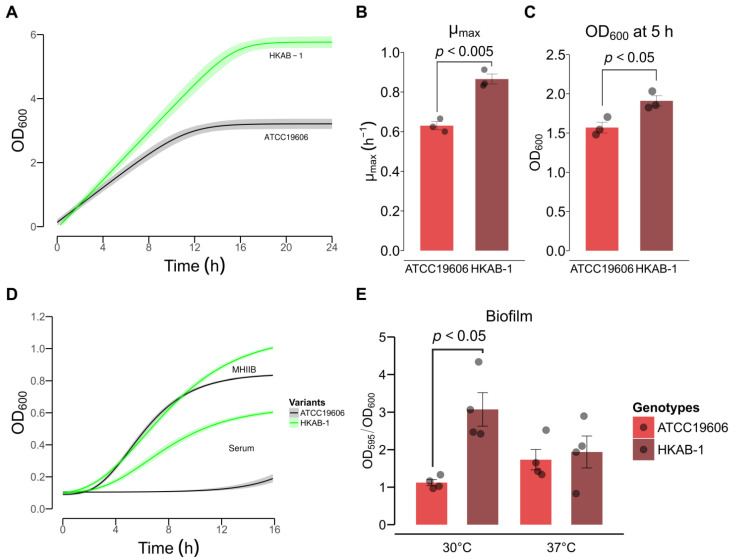
Growth kinetics and biofilm formation of *A. baumannii* laboratory strain ATCC 19606 and clinical isolate HKAB-1. (**A**) Growth curves of *A. baumannii* strains were fitted using the Baranyi and Roberts growth model without a lag phase. Shaded areas represent the 95% confidence intervals derived from bootstrapping. (**B**) Maximum specific growth rates (μ_max_) calculated from the fitted growth model. Statistical analysis was performed using an unpaired Student’s *t*-test. (**C**) Optical density at 600 nm (OD_600_) measured at the 5 h time point. Statistical analysis was performed using an unpaired Student’s *t*-test. (**D**) Growth curves of *A. baumannii* strains grown in MH2B and HKAB-1 strain grown in bovine serum albumin (serum) were fitted using the modified Gompertz growth model, while growth curve of ATCC 19606 grown in bovine serum albumin was fitted using the Baranyi and Roberts growth model without y_max_. Shaded areas represent the 95% confidence intervals derived from bootstrapping. (**E**) Quantification of biofilm biomass formed by *A. baumannii* strains at 30 °C and 37 °C, stained with crystal violet. Data represent individual biological replicates (black dots) and the mean ± SEM from at least three independent biological experiments. Statistical analysis was performed using an unpaired Student’s *t*-test with Welch’s correction for unequal variances where applicable.

**Figure 2 biology-14-01201-f002:**
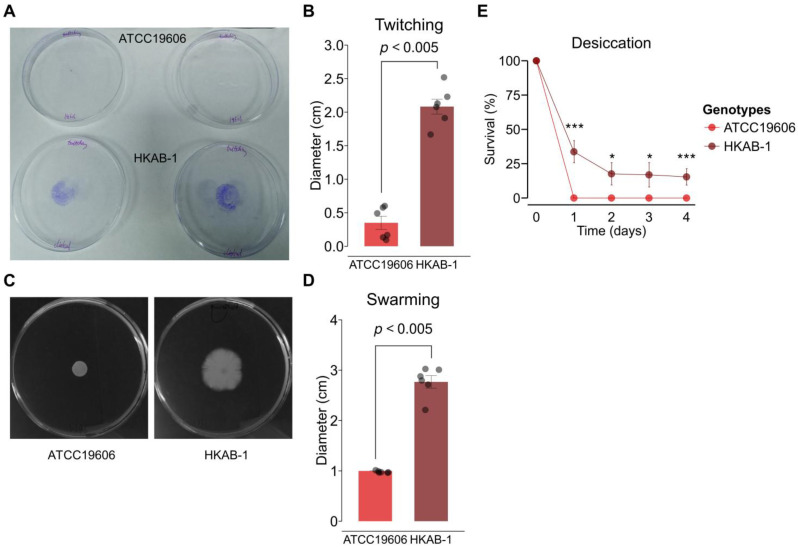
Desiccation tolerance and motility phenotypes of A. *baumannii* laboratory strain ATCC 19606 and clinical isolate HKAB-1. (**A**,**B**) Twitching motility was assessed using polystyrene petri dishes by inoculating bacterial cells at the agar–plastic interphase. The twitching zones were stained with crystal violet after incubation, and the diameters were measured. (**C**,**D**) Swarming motility was evaluated by spotting bacterial cultures onto the surface of semi-solid agar, followed by measurement of the swarming zone diameter. (**B**,**D**) Data represent individual biological replicates (black dots) and the mean ± SEM from at least three independent biological experiments. Statistical analysis was determined using an unpaired Student’s *t*-test. (**E**) Survival of A. *baumannii* ATCC 19606 and HKAB-1 strains under 20–25% relative humidity. Survival is expressed as the percentage of colony-forming units relative to day 0 (set as 100%). Data represent five biological replicates (*n* = 5), with standard errors shown as error bars. Statistical significance was assessed using Tukey’s honestly significant difference test (*, *p* < 0.05; ***, *p* < 0.005).

**Figure 3 biology-14-01201-f003:**
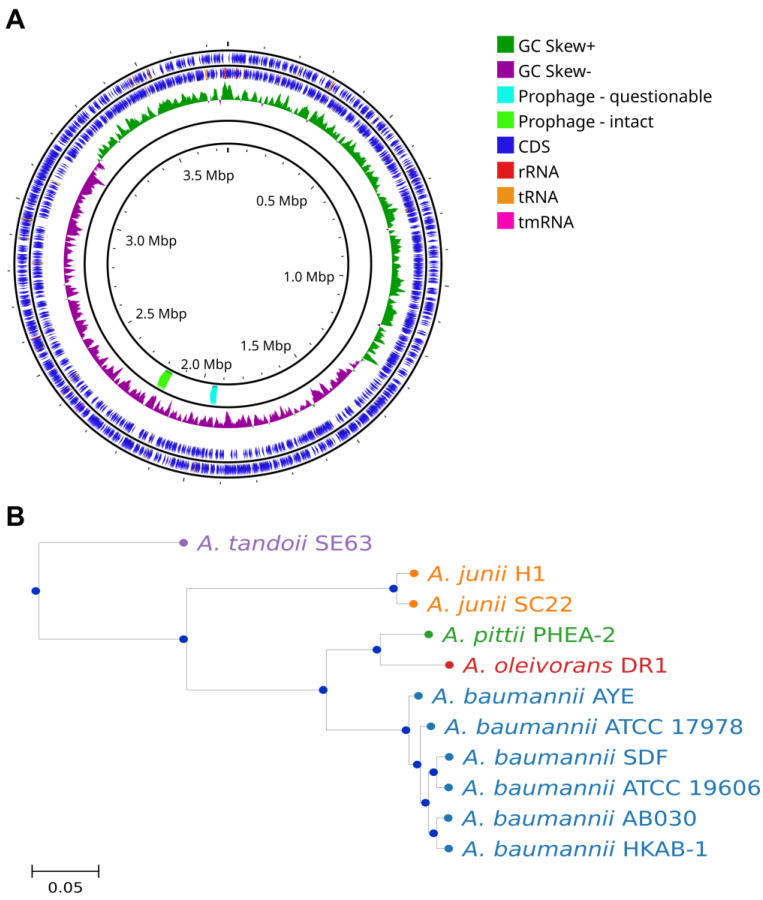
Schematic representation of (**A**) circular chromosome map of the draft genome of HKAB-1 generated through Proksee tool [35], including genome annotation and prophage from PHASTest [34], and (**B**) the core-genome phylogenetic tree of *A. baumannii* HKAB-1 and representative *Acinetobacter* strains. HKAB-1 clusters closely within the *A. baumannii* clade and is clearly separated from non-*baumannii* species such as *A. pittii*, *A. junii*, and *A. tandoii*.

**Figure 4 biology-14-01201-f004:**
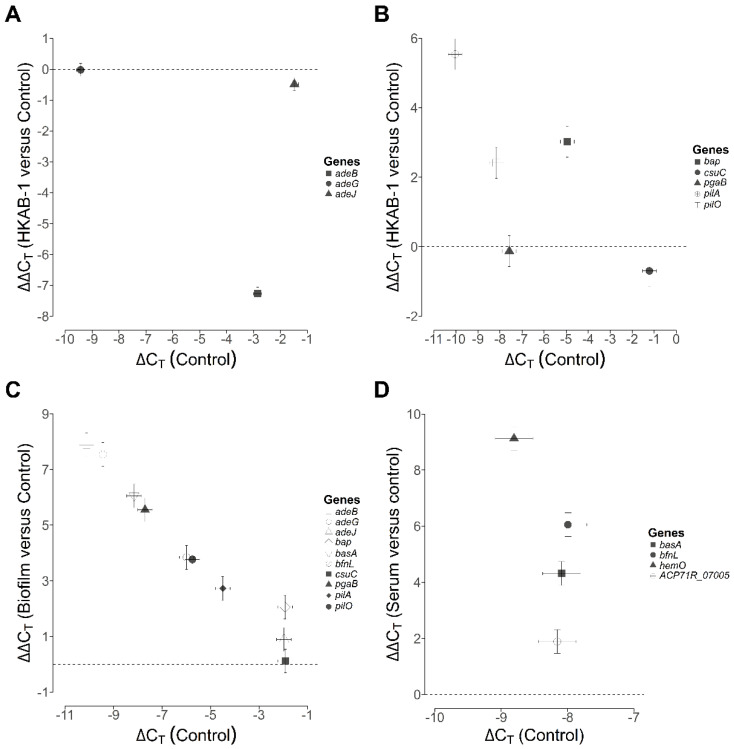
Gene expression analysis of RND efflux pumps and virulence-associated genes in *A. baumannii* HKAB-1 and ATCC 19606. ΔC_T_ values were calculated as C_T(rpoB)_ − C_T(target gene)_. ΔΔC_T_ values represent the mean difference in ΔC_T_ between treatment and control samples (a positive ΔΔC_T_ value indicates induction due to the treatment, while a negative ΔΔC_T_ value indicates repression due to the treatment). The ΔC_T_ of the tested genes in the x-axis represents the calculated ΔC_T_ of the control samples. Points and error bars represent means and SEMs, respectively, of *n* ≥ 3 biological replicates. The SEMs shown were obtained as pooled estimates from the entire dataset. As the sizes of all samples are equal, the bars all have the same lengths. Statistical analysis using Tukey’s HSD test identified several genes with highly significant differences (*p* < 0.005). (**A**) Expression levels of RND genes in HKAB-1. Gene expression of RND genes in ATCC 19606 was used as control. A significant interaction between gene and treatment (HKAB-1) was observed using ANOVA [*F*(3, 12) = 396.70, *p* < 0.001; see Materials and Methods], indicating differential gene regulation among RND genes under the experimental conditions. Notably, *adeB* was significantly downregulated in HKAB-1 (ΔΔC_T_ = −7.27 ± 0.21; fold change = 0.0065). (**B**) Expression levels of biofilm- and motility-associated genes in HKAB-1 (treatment) as compared to ATCC 19606 (control). A significant interaction between gene and treatment was observed using ANOVA [*F*(7, 28) = 34.04, *p* < 0.001; see Materials and Methods], indicating differential gene regulation under the tested conditions. Notably, *bap*, *pilA*, and *pilO* were significantly upregulated in HKAB-1 (*bap*: ΔΔC_T_ = 3.03 ± 0.45; fold change = 8.14; *pilA*: ΔΔC_T_ = 5.54 ± 0.45; fold change = 46.54; *pilO*: ΔΔC_T_ = 2.42 ± 0.45; fold change = 5.34). (**C**) Expression levels of biofilm-associated genes in HKAB-1 biofilm (treatment) as compared to planktonic culture (control). A significant interaction between gene and treatment was observed using ANOVA [*F*(10, 40) = 125.67, *p* < 0.001; see Materials and Methods], indicating differential gene regulation under the tested conditions. Notably, *adeB*, *adeG*, *bap*, *csuC*, *pgaB*, *pilA*, and *pilO* were significantly upregulated in biofilm (*adeB*: ΔΔC_T_ = 7.88 ± 0.43; fold change = 236.19; *adeG*: ΔΔC_T_ = 7.54 ± 0.43; fold change = 185.98; *bap*: ΔΔC_T_ = 2.06 ± 0.43; fold change = 4.17; *basA*: ΔΔC_T_ = 6.05 ± 0.43; fold change = 66.43; *bfnL*: ΔΔC_T_ = 3.84 ± 0.43; fold change = 14.33; *pgaB*: ΔΔC_T_ = 5.55 ± 0.43; fold change = 46.79; *pilA*: ΔΔC_T_ = 2.73 ± 0.43; fold change = 6.62; *pilO*: ΔΔC_T_ = 3.77 ± 0.43; fold change = 13.60). (**D**) Expression levels of genes involved in heme acquisition and siderophore clusters in response to serum (treatment). Gene expression in MH2B was used as control. A significant interaction between gene and treatment was observed using ANOVA [*F*(4, 36) = 199.85, *p* < 0.001; see Materials and Methods], indicating differential gene regulation under the tested conditions. All tested genes were significantly upregulated upon serum exposure (*basA*: ΔΔC_T_ = 4.32 ± 0.42, fold change = 19.96; *bfnL*: ΔΔC_T_ = 6.05 ± 0.42, fold change = 66.29; *hemO*: ΔΔC_T_ = 9.13 ± 0.42, fold change = 558.50; ACP71R_07005: ΔΔC_T_ = 1.89 ± 0.42, fold change = 3.71).

**Table 1 biology-14-01201-t001:** Antimicrobial susceptibility of *A. baumannii* clinical isolate HKAB-1 and ATCC 19606.

MIC (μg/mL)		
HKAB-1	ATCC 19606	Antibiotics
		**Aminoglycosides**
4	8–16	Amikacin
1–2	16–32	Gentamicin
0.5	2–4	Tobramycin
		**Carbapenems**
0.125	0.5	Doripenem
0.125–0.25	0.5–1	Meropenem
		**Cephalosporins**
8–16	16	Cefotaxime
32–64	64–128	Ceftriaxone
32–64	128	Cefuroxime
		**Fluoroquinolones**
0.125	1	Ciprofloxacin
0.125–0.25	0.5	Levofloxacin
		**Macrolides**
2–4	32–64	Azithromycin
16–32	64–128	Clarithromycin
8	16	Erythromycin
		**Penicillins**
16	256	Amoxicillin
512	256	Oxacillin
8–16	32–64	Piperacillin
		**Tetracyclines**
0.25	1–2	Doxycycline
0.125	0.5	Minocycline
2–4	16	Tetracycline
0.125–0.25	4	Tigecycline
		**Other antibiotics**
64–128	64–128	Chloramphenicol
256–512	256	Clindamycin
256	256	Fusidic acid
256–512	256–512	Linezolid
>256	128–256	Nitrofurantoin
>1024	>1024	Fosfomycin
0.25–0.5	0.25–0.5	Polymyxin B
4	2	Rifampicin
1024	1024	Sulfamethoxazole
32–64	64–128	Trimethoprim
256	256–512	Vancomycin

**Table 2 biology-14-01201-t002:** Summary of sequence data and genome features of *A. baumannii* clinical isolate HKAB-1.

Value	Metrics
7,674,099 × 2	Total number of reads sequenced
608.3	Coverage
24	Contig count (≥500 bp)
99.4	Coarse consistency (%)
98.8	Fine consistency (%)
100	Completeness (%)
0.2	Contamination (%)
3,758,367	Genome size (bp)
365,067	Contigs N50 (bp)
4	Contigs L50
38.92	Guanine-cytosine content (%)
3559	Number of genes
3488	Number of coding sequences (CDSs)
63	Number of tRNAs
4	Number of rRNAs

**Table 3 biology-14-01201-t003:** Predicted antimicrobial resistance of *A. baumannii* clinical isolate HKAB-1.

	Antibiotic Resistance		
% Identity	Protein Family	Gene	Locus Tag
*Antibiotic inactivation*			
98.75	β-lactamase	*bla* _OXA-91_	ACP71R_01635
100.00	β-lactamase	*bla* _ADC-50_	ACP71R_02890
99.71	β-lactamase	*bla* _MBL_	ACP71R_06460
98.79	β-lactamase	*bla* _OXA-51_	ACP71R_07365
97.48	β-lactamase	*blaA*	ACP71R_08870
98.61	Aminoglycoside nucleotidyltransferase	*ant(3*″*)-IIa*	ACP71R_14580
	*Efflux pumps and its regulator*		
99.38	Resistance nodulation cell division	*adeK*	ACP71R_01165
99.59	Resistance nodulation cell division	*adeJ*	ACP71R_01170
99.92	Resistance nodulation cell division	*adeI*	ACP71R_01175
99.17	Resistance nodulation cell division	*adeH*	ACP71R_03190
97.23	Resistance nodulation cell division	*adeG*	ACP71R_03195
98.85	Resistance nodulation cell division	*adeF*	ACP71R_03200
99.21	Transcription regulator	*adeL*	ACP71R_03205
99.70	Small multidrug resistance	*abeS*	ACP71R_03230
97.84	Major facilitator superfamily	*amvA*	ACP71R_04480
98.47	Transcription regulator	*adeN*	ACP71R_04740
98.85	Major facilitator superfamily	*abaQ*	ACP71R_05490
97.49	Resistance nodulation cell division	*adeT1*	ACP71R_05965
97.33	Two-component signal transduction system	*adeS*	ACP71R_05975
98.66	Two-component signal transduction system	*adeR*	ACP71R_05980
98.66	Resistance nodulation cell division	*adeA*	ACP71R_05985
98.23	Resistance nodulation cell division	*adeB*	ACP71R_05990
97.75	Major facilitator superfamily	*abaF*	ACP71R_08510
98.89	Transcription regulator	*mexT*	ACP71R_10620
98.52	Multi-antimicrobial extrusion protein	*abeM*	ACP71R_13360
100.00	Major facilitator superfamily	*craA*	ACP71R_16955
99.58	Resistance nodulation cell division	*adeT2*	ACP71R_17230

## Data Availability

The whole-genome sequencing project of *A. baumannii* HKAB-1 has been deposited at NCBI’s GenBank under BioProject accession number PRJNA1255546, BioSample accession number SAMN48147428, and nucleotide accession number CP191205. The raw sequences have been deposited in the SRA under the accession number SRX28891183.

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
