# Peer review of "Genomic and Phenotypic Characterization of a Drug-Susceptible Acinetobacter baumannii Reveals Increased Virulence-Linked Traits and Stress Tolerance"

_biology, 2025, doi:10.3390/biology14091201_

Round 1
Reviewer 1 Report
Comments and Suggestions for Authors
In the manuscript "Genomic and Phenotypic Characterization of a Drug-Susceptible Acinetobacter baumannii Reveals Increased Virulence-linked Traits and Stress Tolerance", Foong et al. characterized the drug-susceptible Acinetobacter baumannii via phynotypic and genomic assays. The study provides important information about new virulence factors of Acinetobacter baumannii strains. The manuscript could be considered for acceptance after major revision. Concerns:
(1) Comparative genome analysis between HKAB-1 and other representative Acinetobacter baumannii strains in the genome database should be performed to show the evolutionary relationship of them.
(2) As shown in Table 1, the strain HKAB-1 exhibits resistance to cephalosporins, penicillins, and some other antibiotics, is it proper to define it as "Drug-Susceptible Acinetobacter baumannii"?
(3) In Figure 1, HKAB-1 had a higher growth rate than the standard strain. What are the possible mechanisms? This should be discsussed in detail.
(4) The gene clusters involved in biofilm formation and swarming motility of HKAB-1 should be analyzed and their functions should be discussed.
(5) The virulence of HKAB-1 should be evaluated by using hemeolysis assays, protease assays and even mouse models.
Reviewer 2 Report
Comments and Suggestions for Authors
The article describes the phenotypic and genomic characterization of a clinical Acinetobacter baumannii isolate (HKAB-1) that displays unexpectedly high virulence-associated traits despite being susceptible to a wide range of antibiotics. The study aims to explore the paradoxical relationship between antimicrobial susceptibility and pathogenic potential by comparing HKAB-1 with the reference strain ATCC19606. Through a combination of growth assays, biofilm formation, motility tests, desiccation tolerance experiments, and whole-genome sequencing, the authors highlight important insights into the mechanisms underlying virulence in non-MDR A. baumannii.
In my opinion, the article is well-written and presents a highly relevant topic. Acinetobacter baumannii is widely recognized as one of the most significant threats in hospital-acquired infections, particularly due to its association with multidrug resistance. The study is experimentally well-designed, employing a broad range of phenotypic and genomic methods. The data are consistent and clearly presented, supported by appropriate statistical analysis. However, before the article can be published, the authors should address the following suggestions:
- The introduction lacks a clear statement of the research hypothesis. Although the hypothesis is briefly outlined in the abstract and discussed in the Discussion section, it is not explicitly or formally presented in the Introduction. I suggest adding a sentence such as "The aim of this study was..." to clearly define the study’s objective at the end of the Introduction.
- The hypothesis of a potential trade-off between antibiotic resistance and virulence is certainly intriguing and relevant. However, this conclusion should be stated with more caution, as the manuscript does not present direct molecular evidence to support it. While the phenotypic assays (e.g., enhanced biofilm formation, motility, and desiccation tolerance) are well-documented, and the genomic analysis reveals the presence of resistance and virulence genes, the study does not include functional validation of gene expression or regulatory mechanisms (e.g., RNA expression analysis or proteomics). Therefore, I recommend rephrasing the relevant statements to reflect that this trade-off is a hypothesis or observation, rather than a confirmed mechanistic link. For example, you may write: “Our findings suggest a potential trade-off…” or “The observed phenotype may indicate…”, and acknowledge the need for further studies to explore the molecular basis of this relationship.
- Genotype does not always translate directly into phenotype. Throughout the manuscript, the presence of certain genes (e.g., adeB) is often interpreted as evidence of active resistance or virulence mechanisms. However, without functional validation, such as gene expression or protein activity assays, this interpretation may be overstated. I recommend adopting a more nuanced phrasing — for example, using terms like “potential activity” or “putatively functional” — to reflect that gene presence alone does not confirm functional expression or contribution to phenotype.
- One of the main limitations of the manuscript is the complete lack of clinical data regarding the patient, apart from the mention of left-sided heart failure. This omission weakens the clinical context of the infection and reduces the translational value of the findings. Although the study is retrospective, it would greatly enhance the relevance and impact of the work if the authors could include even brief information about the infection course, duration of hospitalization, clinical management, and treatment outcomes. If such data are available, I strongly encourage the authors to incorporate them into the current manuscript.
- The study lacks RNA-seq or RT-qPCR data, which could have supported the conclusions regarding the expression of virulence and resistance genes (e.g., ade, bap, hemO). Including such functional data would strengthen the link between genotype and phenotype. I recommend adding a brief comment on this limitation in the Discussion section.
Round 2
Reviewer 1 Report
Comments and Suggestions for Authors
The authors have carefully revised the manuscript. It is acceptable.